# Copper Uptake and Accumulation, Ultra-Structural Alteration, and Bast Fibre Yield and Quality of Fibrous Jute (*Corchorus capsularis* L.) Plants Grown under Two Different Soils of China

**DOI:** 10.3390/plants9030404

**Published:** 2020-03-24

**Authors:** Muhammad Hamzah Saleem, Shafaqat Ali, Sana Irshad, Muhammad Hussaan, Muhammad Rizwan, Muhammad Shoaib Rana, Abeer Hashem, Elsayed Fathi Abd_Allah, Parvaiz Ahmad

**Affiliations:** 1MOA Key Laboratory of Crop Ecophysiology and Farming System in the Middle Reaches of the Yangtze River, College of Plant Science and Technology, Huazhong Agricultural University, Wuhan 430070, China; saleemhamza312@webmail.hzau.edu.cn; 2Department of Environmental Sciences and Engineering, Government College University Allama Iqbal Road, 38000 Faisalabad, Pakistan; mrazi1532@yahoo.com; 3Department of Biological Sciences and Technology, China Medical University, Taichung 40402, Taiwan; 4School of Environmental Studies, China University of Geosciences, Wuhan 430070, China; sanairshad55@gmail.com; 5Department of Botany, Government College University, Allama Iqbal Road, 38000 Faisalabad, Pakistan; mhussaan7866@gmail.com; 6Key Laboratory of Arable Land Conservation (Middle and Lower Reaches of Yangtze River), Ministry of Agriculture, Microelements Research Center, College of Resource and Environment, Huazhong Agricultural University, Wuhan 430070, China; muhammadshoaib1555@gmail.com; 7Botany and Microbiology Department, College of Science, King Saud University, P.O. Box. 2460, Riyadh 11451, Saudi Arabia; habeer@ksu.edu.sa; 8Mycology and Plant Disease Survey Department, Plant Pathology Research Institute, ARC, Giza 12511, Egypt; 9Plant Production Department, College of Food and Agricultural Sciences, King Saud University, P.O. Box. 2460, Riyadh 11451, Saudi Arabia; eabdallah@ksu.edu.sa; 10Department of Botany, S.P. College, Srinagar 190001, Jammu and Kashmir, India

**Keywords:** *Corchorus capsularis*, copper contaminated soil, gaseous exchange attributes, phytoremediation, antioxidants

## Abstract

Copper (Cu) is an essential heavy metal for plants, but high Cu concentration in the soil causes phytotoxicity. Some plants, however, possess a system that can overcome Cu toxicity, such as Cu localization, and an active antioxidant defence system to reduce oxidative damage induced by high Cu concentration. The present study was conducted to explore the phytoremediation potential, morpho-physiological traits, antioxidant capacity, and fibre quality of jute (*Corchorus capsularis*) grown in a mixture of Cu-contaminated soil and natural soil at ratios of 0:1 (control), 1:0, 1:1, 1:2 and 1:4. Our results showed that high Cu concentration in the soil decreased plant growth, plant biomass, chlorophyll content, gaseous exchange, and fibre yield while increasing reactive oxygen species (ROS), which indicated oxidative stress induced by high Cu concentration in the soil. Antioxidant enzymes, such as superoxidase dismutase (SOD), peroxidase (POD), catalase (CAT) and ascorbate peroxidase (APX) scavenge ROS in plant cells/tissues. Furthermore, high Cu concentration did not significantly worsen the fibre quality of *C. capsularis*, and this plant *was* able to accumulate a large amount of Cu, with higher Cu accumulation in its shoots than in its roots. Transmission electron microscopy (TEM) revealed that Cu toxicity affected different organelles of *C. capsularis*, with the chloroplast as the most affected organelle. On the basis of these results, we concluded that high Cu concentration was toxic to *C. capsularis*, reducing crop yield and plant productivity, but showing little effect on plant fibre yield. Hence, *C. capsularis*, as a fibrous crop, can accumulate a high concentration of Cu when grown in Cu-contaminated sites.

## 1. Introduction

Heavy metal accumulation has become a severe problem in China and other countries owing to many documented cases of sewage sludge, industry, mining and uses of pesticides and herbicides [1,2,3,4,5]. There are many toxic pollutants in the soil commonly, copper, arsenic, cadmium, zinc, and iron, are some toxic heavy metals, particularly in the capacities with high pollution rate. Among different heavy metals, Cu is an essential heavy metal required by plants in minute quantity for many different biological processes [6,7,8]. Cu is a micronutrient, and excess Cu concentration is toxic to plants. Excess Cu in the soil causes alterations in DNA and cell membrane integrity, ultimately reducing plant growth and biomass [9,10,11,12]. Phytotoxicity of Cu causes membrane damage through binding of Cu to the sulfhydryl groups of membrane proteins. Furthermore, normal concentration of Cu is nontoxic to plants, whereas Cu toxicity directly disturbs the structure of chloroplast, mitochondrial apparatus, nucleus, and other cellular organelles [6,7,13,14]. Excess Cu causes damage to the cellular membrane organelles by increasing ROS in plant cells and tissues, causing peroxidation of substances on the cell surface [11,15,16,17]. ROS, which are toxic and remove by the activities of antioxidant compounds [18,19,20,21,22,23]. Previously, in many different crops, activities of high antioxidants were found such as in *Boehmeria nivea* [15], *Phyllostachys pubescens* [13] and *Brassica napus* [24] under Cu stress.

Recently, phytoextraction has become increasingly popular owing to its eco-friendly, cheap, and scientifically accepted properties; moreover, it can be easily applied on-site and easily monitored [16,25,26,27]. different plant species have been studied for phytoremediation of various metal contaminants [13,28,29,30]. Among these plants, fibrous crops such as *Linum usitatissimum* [31], *Hibiscus cannabinus* [32] and *Boehmeria nivea* [15] have been used for phytoextraction of Cd, Cr and Cu. Jute (*C. capsularis* L.) is considered as potential candidate against stress condition than other fibrous crop, possibly owing to its specific biochemical and physiological responses [33,34]. *C. capsularis* has been cultivated for its fibre worldwide, especially in Bangladesh, India and China [35,36]. Its fibre is a type of natural bast fibre with low-cost and, presently, the maximum production volume. *C. capsularis* fibre-reinforced composites have been utilised in the automobile, footwear, construction, furniture, and different industries. It is also applied for the production of false ceilings, different compounds chlorophyll papers, toilet tiles, and tables [35,37,38,39]. In addition to this, *C. capsularis* is an excellent candidate for the phytoremediation of different heavy metals which we have discussed in detail in our review literature [40].

The present study was conducted to explore the potential of phytoremediation by *C. capsularis* in Cu-contaminated soil with regards to its growth, chlorophyll content, gaseous exchange, antioxidant capacity, fibre yield and quality and ultra-structural changes in its chloroplasts. There are a few kinds of literature available on the growth and phytoremediation potential of *C. capsularis* [34,41,42,43]. Still, no study has been found on the fibre quality and ultra-structural changes in the chloroplast of *C. capsularis* grown in a metal-contaminated soil. To the best of our knowledge, this study is the first of such studies. The results of the present study will advance our knowledge on the (i) phytoremediation potential; (ii) growth, biomass and antioxidant capacity; (iii) as well as fibre quality and ultra-structural changes in the chloroplast of *C. capsularis* grown in a mixture of Cu-contaminated soil and natural soil.

## 2. Results and Discussion

### 2.1. Effect of Cu on Plant Morphology and Composition

In the present study, *C. capsularis* was used to investigate Cu accumulation and physiological responses against metal stress. The effect of various treatments of Cu in the soil on growth of the plant was also examined (Table 1). Table 1 shows that elevating Cu level in the soil significantly (*p* > 0.05) reduced the height as well as total fresh and dry weight of *C. capsularis*. The maximum plant height (241 cm) as well as fresh (159 g) and dry biomass (58 g) were observed in control, but increasing Cu contents in the soil decreased plant growth and development. In contrast, the maximum reductions in plant height (23.3%), FW (33.9%), and dry biomass (41.3%) were observed in T4 compared to those in control. The decline in plant growth and biomass was probably caused by inhibited cell elongation and division due to toxic contents of Cu in the soil [44,45]. In many previous studies, high Cu contents in the soil has been shown to reduce plant growth and biomass [9,13,26]. However, previously we reported that jute could tolerate the Cu level up to 300 mg kg^−1^ and Cu contents more than 300 mg kg^−1^ in the soil caused decreased in plant growth [46]. This might be due to the high level of Cu accumulated in the cells, which affects the plant growth and development [7,17]. Another reason behind this mechanism is also due to the reduction in essential elements in the soil and an increased in Cu contents in plant cells, causing chlorotic symptoms [47].

### 2.2. Effect of Cu Chlorophyll Content and Gaseous Exchange

The most common impact of Cu stress is a decrease in photosynthetic pigments in the leaves. Moreover, a decrease in photosynthesis activity and alteration in the ultrastructure of the chloroplast is directly linked with photosynthetic pigments in the leaves [11,15]. Total chlorophyll (chl a and b) is also affected by toxic Cu level in the soil. Photosynthetic pigments in the leaves of *C. capsularis* is shown in Table 1. Data showed that the highest total chlorophyll was decreased by 44.8% in T4 plants, followed by that in T3 (31%) and T2 (17.2%), compared to the control. Previous studies showed that a high content of Cu in the soil causes a decrease in photosynthetic pigments [16,48]. Although Cu phytotoxicity affects both photosystems in plants, PS II is more sensitive than PS I to Cu stress [7,24,49].

The possible reason behind this mechanism of reduced contents of chlorophyll is the damage in membrane bounded organelles of the cell i.e., thylakoid membrane and Cu-induced interference with chlorophyll organisation [6,48,50]. The gaseous exchange was also affected by toxic level of Cu in the soil. *Pn*, *gs*, *Ts*, and *Ci* are presented in Figure 1. Our results showed that elevating concentration of metal reduced *Pn*, *gs*, *Ts* and *Ci* compared with those of the plants grown in Cu-free soil. The maximum reduction of *Pn*, *gs*, *Ts* and *Ci* in T4 plants was 55.6%, 88.2%, 58.6% and 13.2%, respectively, compared to those in control. *Pn*, *gs* and *Ts* significantly (*p* > 0.05) decreased owing to excess Cu (T4), whereas *Ci* showed a non-significant decrease; however, T4 significantly (*p* > 0.05) decreased *Ci*. The decreased activities of photosynthetic apparatus might be due to excess Cu in the soil, which affects PS II machinery, resulting in lower photosynthetic electron transport activities [9,26]. Similar results were showed by Rehman et al. [9], who found that toxic amount of metal reduced photochemistry of *B. nivea*.

### 2.3. Effect of Cu on Fibre Yield and Quality

The fibre yield of *C. capsularis* is presented in Table 1. Fibre yield and quality was decreased by the toxic level of Cu in the soil (Table 1, Figure 2). Fibre yield (raw fibre yield and degummed fibre yield) decreased by 32.6% and 64%, respectively, in T4 plants compared to that in control. It was also noticed that maximum fibre diameter (50%), fibre elongation rate (60.6%) and fibre breaking strength (70.9%) were decreased in the plants which were grown in T1 plants compared to the control. In our study, it was noticed that Cu concentration caused a different effect on fibre yield and quality. Similar results were shown by Linger et al. [51] in their study of industrial hemp (*Cannabis sativa*) under different metal stress, in which heavy metal was shown to slightly decrease the fibre quality of *C. sativa*. Although, detailed mechanism of fibrous crop and their yield was studied by Ullah et al. [52] and they highlighted that plant with long stem and huge biomass may provide a good and quality fibrous yield. However, there are very few literatures available on fibre quality under heavy metal stress, it is generally known that Cu toxicity in the soil exerts very slight effect on *C. capsularis* as this plant has a large biomass and thick fibre production compared to other fibrous crop. In our previous study, we also determined that the phytotoxicity of Cu directly affected the yield and quality of different jute vars [53].

### 2.4. Effect of Cu on Oxidative Stress and Antioxidative Activities

Stress condition cause high production of ROS in the cell/tissue and elimination under normal growth in plants, thus promoting ROS production, oxidative stress, and destroy the structure of many membrane bounded organelles [9,17]. Many previous studies have noted that high Cu concentration in the soil increases lipid peroxidation, which is toxic to cellular organelles of plants [13,20]. In our study, increased malondialdehyde (MDA) content under high Cu concentration indicated that Cu induced oxidative damage in the cells or tissues of *C. capsularis*, which also increased H_2_O_2_ initiation and electrolyte leakage (EL). Furthermore, increasing MDA content in the leaves indicated oxidative stress due to the high toxicity of Cu in the soil. Results of MDA, H_2_O_2_ content and EL are presented in Figure 3. The results revealed that the maximum MDA and H_2_O_2_ content enhanced by 408.6% and 1135.9%, respectively, in T4 plants compared to that in control, whereas EL increased by 900% compared to that in control. Similar trends that increasing Cu contents in the soil significantly enhanced in MDA, H_2_O_2_ and EL were observed in all treatments. ROS production is a common feature of Cu stress in plants. For example, high concentration of Cu in the soil/medium caused oxidative stress in the leaves of rapeseed and wheat [16,54]. Alternatively, binding of Cu to the cell wall could directly alteration its elasticity or by replacing Ca. The increases in lipid peroxidation, H_2_O_2_ content, EL and oxidative damage might be due to the toxic level of metal in the soil, which influence stress and affect many different processes in different plant species [20,28,55].

The high concentration of ROS is toxic to plants, but plants have specialised defence systems that can reduce ROS toxicity [19,56]. In this experiment, the enzymatic activities of antioxidants were measured under toxic level of metal in the soil (Figure 4). It was noticed that elevating level of Cu in the soil increased antioxidant activities compared to that in control (Figure 4). The highest antioxidant activities were found in T4, and increasing the amount of natural soil in Cu-contaminated soil significantly decreased the antioxidant activities. Increasing antioxidant activities under toxic level of metal in the soil indicated Cu stress. These results depicted that antioxidants activities were increased by 677.8%, 490.9%, 223% and 479% in T4 plants compared to those in control. However, antioxidant activities decreased significantly with the addition of natural soil in the Cu-contaminated soil compared to those in control. Increasing antioxidant activities under stress conditions have been observed in numerous studies [13,15,17,57]. The elevating activities of antioxidants under stressful condition possibly due to the changes in some gene functions and different proteins in the plant species. Previously, elevating levels of Cu in the medium significantly enhanced antioxidant activities in *moso bamboo*, *Orzya sativa*, and *Eclipta alba* [13,58,59].

Transportation and accumulation of metal in the plant parts depend on metal supply, growth conditions, and plant species. *C. capsularis* is considered a hyperaccumulator specie that is able to absorb a huge concentration of Cu in its shoot parts than in its root parts. Ahmad and Slima [33] studied *C. capsularis* grown in water sludge-contaminated soil and found that most of heavy metals in their shoot parts than to the root parts. In our previous studies, we also noticed that jute is a hyperaccumulator species and is able to uptake a large amount of heavy metals in their shoots than in roots of the plants [40,47,53]. Similar findings were showed by Uddin et al. [60] and Uddin Nizam et al. [42], who studied jute grown in Pb and As-contaminated soil. Cu accumulation results revealed that increasing Cu concentration in the soil significantly (*p* > 0.05) enhanced Cu contents in the roots and shoots of *C. capsularis* compared to that in the plants grown in Cu-free soil (Table 2). Furthermore, the highest Cu contents was found in the above-ground parts, whereas the minimum Cu concentration was absorbed in the below-ground parts of the plant (Table 2). The maximum Cu contents accumulated in the shoots was the highest in T4 (1965 ± 8) followed by T3 (1020 ± 5) and T2 (525 ± 8) plants compared to that in control. The maximum Cu concentration in the roots was the highest in T4 (1275 ± 10) followed by T3 (700 ± 9) and T2 (203 ± 8) plants compared to that in control. However, the ratio between Cu concentration in shoots to the roots was in between 2–3.4 in all treatments of the present study. *C. capsularis* has been used for phytoextraction of various heavy metals in numerous studies [34,41,43,60,61]. 

### 2.5. Correlation 

A correlation graph was carried to study the differences in growth parameters and Cu accumulation in *C. capsularis* (Figure 5). Cu contents in the roots of the plant were positively linked with Cu contents in the harvestable parts of the plants, but was negatively correlated with plant growth, biomass, fibre yield and quality and gaseous exchange. Similarly, antioxidant enzyme activities and oxidative stress were positively correlated with each other as well as Cu accumulation in the belowground parts and shoots, but were negatively linked with plant growth, fibre quality and gaseous exchange. Plant height was positively correlated with plant biomass, fibre quality and gaseous exchange, but was negatively correlated with Cu accumulation, antioxidant enzyme activities and oxidative stress. This correlation analysis revealed a close relationship between different parameters of *C. capsularis* plants.

### 2.6. Ultra-Structural Alteration in the Chloroplast

The chloroplast, nucleus, mitochondria and ribosomes are the key cellular organelles in most life activities [44,45]. The toxic effect of Cu is related to many physiological and biological processes, which suggested disabilities in the structural level of the cell. Changing in the cellular structure of the cell is due to the toxic level of metal in the soil [13,44]. Studies of these alterations at the cellular level have contributed to identify the affected damage areas as well as the disturbance in the plant parts and its composition. It was observed that low concentration of Cu causes a little effect on the structure of the chloroplast, nucleus, mitochondria and ribosomes, i.e., T1 and T2 contained low Cu concentration in the soil and affected little alteration in the structure of the chloroplast (Figure 6). The chloroplast is the main site of cellular injury under Cu stress [6,13]. In T1 plants, the organelles of the leaves were damaged heavily by exposure to excess Cu in the soil. It was also observed that a large number of chloroplast particles was accumulated inside the cell wall and outside the chloroplast in T1 plants. At a low concentration, Cu regulates cell structure, whereas, at a high concentration, it damages the structural integrity of plant cell membrane [14]. Although there is no previous study of *C. capsularis* to study ultra-structure alteration of chloroplast under Cu-stress, we demonstrated in pot experiment that Cu toxicity disturbs cellular organelles in *C. capsularis* plants while fertilization of P improved membrane-bounded structures which were investigated with TEM analysis [53].

## 3. Material and Methods

### 3.1. Plant Growth and Experimental Treatment

A pot experiment was conducted in a glasshouse environment under controlled conditions at Huazhong Agricultural University, Wuhan, China (114.20′ E, 30.28′ N) from 15th March 2019. The seeds were surface-sterilised using 3.5% sodium hypochlorite for 20 min and rinsed three times with distilled water. The seeds of jute (*Corchorus capsularis* L.) used in this experiment were of C-3 variety, which is a type of white jute that originated from Bangladesh. Uncontaminated soil was collected from experimental stations of Huazhong Agricultural University, and Cu-contaminated soil was collected from a Cu mining area of Baisha Village, DaYe County, Hubei, China (115.20′E, 29.85′N) at a depth of 0–25 cm. Natural soil was mixed with Cu-contaminated soil to reduce Cu concentration in the soil as follows: Ck (soil without addition of Cu-contaminated soil), T1 (Cu-contaminated soil is mixed with natural soil by 1:4), T2 (Cu-contaminated soil is mixed with natural soil by 1:2), T3 (Cu-contaminated soil is mixed with natural soil by 1:1) and T4 (Cu-contaminated soil is mixed with natural soil by 1:0). The physicochemical properties of the soils used for the pot experiment were as follows: Ck: pH 6.35 ± 0.03, EC 205 ± 2, CEC 11 ± 0.15; T4: pH 6.90 ± 0.01, EC 273 ± 2, CEC 15.4 ± 0.1; T3: pH 66.76 ± 0.04, EC 250 ± 5, CEC 14.7 ± 0.1; T2, pH 6.56 ± 0.02, EC 232 ± 4, CEC 12.5 ± 0.1; and T1: pH 6.45 ± 0.04, EC 217 ± 2, CEC 11.9 ± 0.1. EC (μscm^−1^) is electrical conductivity, and CEC (cmol kg^−1^) is cation exchange capacity. Other physicochemical properties in natural soil are as follow: 23.16 g kg^−1^ organic matter, 60 mg kg^−1^ exchangeable K, 0.18 g kg^−1^ total P, 40 g kg^−1^ total N and in Cu-contaminated soil are as follow: 3.96 g kg^−1^ organic matter, 12.25 g kg^−1^ exchangeable K, 1.97 g kg^−1^ total P, 0.16 g kg^−1^ total N.

In all treatment groups, Cu concentration was determined before the start of the pot experiment, and the results are as follows: Ck, 34 mg kg^−1^ Cu; T1, 325 mg kg^−1^ Cu; T2, 575 mg kg^−1^ Cu; T3, 1315 mg kg^−1^ Cu; and T4, 2153 mg kg^−1^ Cu. We have already used the same treatments in our previous study on flax [62]. After the two soils were mixed, pots were equilibrated for 2 weeks by one cycle of saturation with distilled water and air-drying. All pots (height × width = 30 cm × 40 cm) were filled with 16 kg of soil and placed in a glasshouse, where they received natural light with day/night temperatures of 35/30 °C and day/night humidity of 70/90%. Each treatment was arranged in a completely randomised design with six replications and three plants in each pot. Weeding, irrigation with Cu-free water and other necessary intercultural operations were performed when needed. Nitrogen, phosphorus and potassium were applied as recommended by Islam et al. [63]. All plants were harvested at 120 days after sowing (DAS) for the examination of morphological traits. All chemicals used were of analytical grade and procured from Sinopharm Chemical Reagent Co., Ltd.

### 3.2. Sampling and Data Collection

All plants were harvested in the second week of July 2019. Fully functional leaves (the fifth from the top) were collected at 60 DAS for evaluation of enzymes and antioxidants. The leaves were washed with distilled water, immediately placed in liquid nitrogen, and stored then in the refrigerator (-80 °C) for further analysis. Morphological traits, such as plant height, fresh plant weight, and plant dry weight was measured after harvesting at 120 DAS. Five uniform plants were randomly selected for trait measurement. Plant height, defined as the total length of the plant (i.e., from the tip of the roots to the uppermost part of the leaves), was measured by using a measuring scale. Plant fresh weight was measured by measuring the total weight of the plant, including root and shoot weight, using a digital balance. For measuring plant dry weight, plant samples were oven-dried at 105 °C for 1 h, followed by at 65 °C for 72 h until the weight was uniform. Root samples were immersed in 20 mM Na_2_EDTA for 15–20 min to remove Cu that adhered to the root surface. Next, the roots were washed thrice with distilled water, washed once with de-ionised water and dried for further analysis [64].

### 3.3. Leaf Chlorophyll and Leaf Photosynthesis

Leaves were collected at 60 DAS for determination of chlorophyll content. For chlorophyll content analysis, 0.1 g of fresh leaf sample was extracted with 8 mL of 95% acetone for 24 h at 4 °C in the dark. The absorbance was measured by a spectrophotometer (UV-2550; Shimadzu, Kyoto, Japan) at 646.6, 663.6 and 450 nm. Chlorophyll content was calculated by the standard method of Arnon [65].

At the same days, gaseous exchange was also measured. Net photosynthesis (*Pn*), leaf stomatal conductance (*gs*), transpiration rate (*Ts*), and intercellular carbon dioxide concentration (*Ci*) were measured from three different plants in each treatment group. Measurements were conducted between 11:30 and 13:30 on days with a clear sky. Rates of leaf *Pn*, *gs, Ts*, and *Ci* were measured with an LI-COR gas-exchange system (LI-6400; LI-COR Biosciences, Lincoln, NE, USA) with a red-blue LED light source on the leaf chamber. In the LI-COR cuvette, CO_2_ concentration was set as 380 mmol mol^−1^ and LED light intensity was set at 1000 mmol m^−2^ s^−1^, which is the average saturation intensity for photosynthesis in *C. capsularis* [66].

### 3.4. Fibre Yield and Fibre Quality

The fibre layer of each stem was decorticated (peeled from the pith), the epidermis was removed, and raw fibres were weighed for calculation of fibre yield. Next, 20 g of decorticated fibre was boiled for 1 h in an Erlenmeyer flask containing 100 mL of degumming solution (1 g NaOH and 0.05 g EDTA). The degummed fibres were bleached with 2% H_2_O_2_ and 0.1% Tween-80 for 1 h at 94 °C in a water bath, washed with distilled water, dried and combed. Fibre diameter (µm) was measured using a digital fibre fineness tester (Model No. YG002C; Changzhou, China) connected to an optical microscope. Fibre breaking strength (centi newtons, cN) and elongation rate (%) were determined using a fibre strength tester (YG004; Nantong Hongda Experiment Instruments, Qidong, China), following the Chinese National Standards (GB 5882–86)

### 3.5. Oxidative Stress and Antioxidative Activities

The degree of lipid peroxidation was evaluated as malondialdehyde (MDA) content. Briefly, 0.1 g of frozen leaves were ground at 4 °C in a mortar with 25 mL of 50 mM phosphate buffer solution (pH 7.8) containing 1% polyethene pyrrole. The homogenate was centrifuged at 10,000 × *g* at 4 °C for 15 min. The mixtures were heated at 100 °C for 15–30 min and then quickly cooled in an ice bath. The absorbance of the supernatant was recorded by using a spectrophotometer (xMark™ microplate absorbance spectrophotometer; Bio-Rad, United States) at wavelengths of 532, 600 and 450 nm. Lipid peroxidation was expressed as l mol g^−1^ using the following formula: 6.45 (A532-A600)-0.56 A450. Lipid peroxidation was measured using a method previously published by Health and Packer [67] and Dionisio-Sese and Tobita [68] respectively while H_2_O_2_ inititation were measured using Assay Kit.

To evaluate enzyme activities, fresh leaves (0.5 g) were homogenised in liquid nitrogen and 5 mL of 50 mmol sodium phosphate buffer (pH 7.0) including 0.5 mmol EDTA and 0.15 mol NaCl. The homogenate was centrifuged at 12,000 × *g* for 10 min at 4 °C, and the supernatant was used for measurement of SOD and POD activities. SOD activity was assayed in 3 mL reaction mixture containing 50 mM sodium phosphate buffer (pH 7), 56 mM nitroblue tetrazolium, 1.17 mM ribolavin, 10 mM methionine and 100 μL enzyme extract. Finally, the sample was measured by using a spectrophotometer (xMark™ microplate absorbance spectrophotometer; Bio-Rad). Enzyme activity was measured using a method by Chen and Pan [69] and expressed as U g^−1^ FW.

POD activity in the leaves was estimated using the method of Sakharov and Ardila [70] using guaiacol as the substrate. A reaction mixture (3 mL) containing 0.05 mL of enzyme extract, 2.75 mL of 50 mM phosphate buffer (pH 7.0), 0.1 mL of 1% H_2_O_2_ and 0.1 mL of 4% guaiacol solution was prepared. Increases in the absorbance at 470 nm because of guaiacol oxidation was recorded for 2 min. One unit of enzyme activity was defined as the amount of the enzyme.

The enzymatic activity of CAT was measured by the method of Aebi [71], and expressed as U g^−1^ FW.

Ascorbate peroxidase activity was measured according to a method by Nakano and Asada [72], and expressed in U g^−1^ FW.

### 3.6. Cu Determination

Dried root and shoot (leaves, stems, and fibres) samples were ground in a stainless-steel mill and passed through a 0.1-mm nylon sieve for Cu analysis. Briefly, 0.1 g of dried sample was digested in HNO_3_/HClO_4_ (4:1) solution. The digested solution was washed in 25-mL flasks and diluted in de-ionised water until reaching the final volume of 25-mL. The supernatant was passed through a 0.45-μm filter paper and determined using a Perkin-Elmer 3100 Atomic Absorption Spectrophotometer, which calibrated with standard solutions containing known concentrations of each element. 

### 3.7. Transmission Electron Microscopy

For TEM, leaf samples were collected at 60 DAS and placed in liquid nitrogen. Small sections of the leaves (1−3 mm in length) were fixed in 4% glutaraldehyde (v/v) in 0.2 mol/L SPB (sodium phosphate buffer, pH 7.2) for 6−8 h and post-fixed in 1% OsO_4_ for 1 h, then in 0.2 mol/L SPB (pH 7.2) for 1−2 h. Samples were dehydrated in a graded ethanol series (50%, 60%, 70%, 80%, 90%, 95% and 100%) followed by acetone, filtered and embedded in Spurr’s resin. Ultra-thin sections (80 nm) were prepared and mounted on copper grids for observation under a Hitachi 500 electron microscope at an accelerating voltage of 60.0 kV or 80.0 kV. 

### 3.8. Statistical Analysis

Experimental data were analysed following the method outlined by Gomez and Gomez [73], and Duncan’s multiple range test was performed to verify the significance of differences. Normality test showed that all plant- or soil-related data were approximately normally distributed. Thus, differences between treatments were determined using analysis of variance, and the least significant difference test (*p* ≤ 0.05) was used for multiple comparisons between treatment means. Data were tested with one-way analysis of variance followed by LSD tests using Statistix 8.1. The graphical presentation was carried out using SigmaPlot 12.5 and R 3.4.1.

## 4. Conclusions

Based on these results, we concluded that *C. capsularis* could grow in Cu-contaminated soil owing to its active defence system. However, high Cu concentration decreased the growth and biomass of *C. capsularis*. Furthermore, Cu induced oxidative damage in the leaves of *C. capsularis* and ultra-structural disturbance in various cellular organelles, especially the chloroplast. However, *C. capsularis* was able to accumulate a large amount of Cu in their above-ground parts and grow as a fibrous crop (as a minimal effect of Cu on the fibre was observed) in Cu-contaminated soil. However, further field experiment is recommended to advance our understanding of the phytoremediation behaviour of the plant.

## Figures and Tables

**Figure 1 plants-09-00404-f001:**
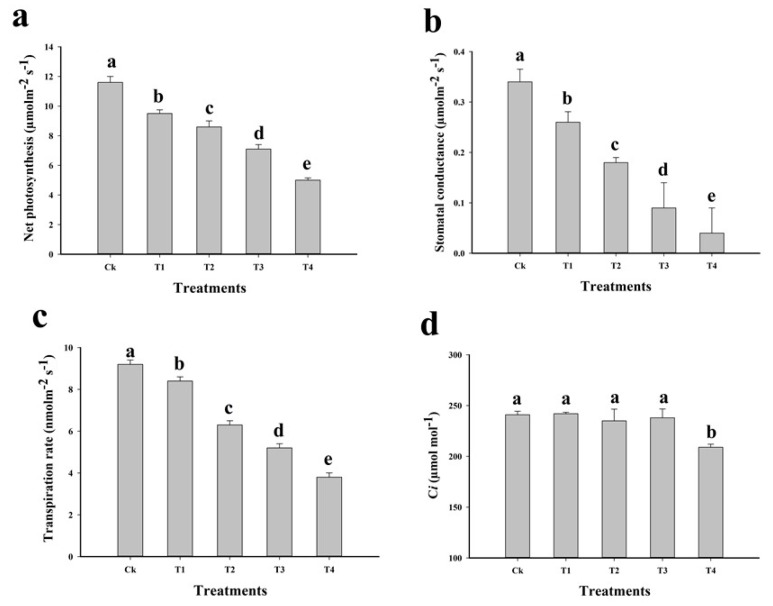
Effect of different levels of Cu-contaminated soil mixed with natural soil on net photosynthesis (**a**), stomatal conductance (**b**), transpiration rate (**c**) and intercellular CO_2_ in the leaves of *C. capsularis*. Values in the figures are just one harvest. Bars sharing different lowercase letters denote significant statistical differences among the treatments at *p* < 0.05 according to least significant difference (LSD) test. Data shown are the average of three replications (*n* = 3). Ck (soil without addition of Cu-contaminated soil), T1 (Cu-contaminated soil is mixed with natural soil by 1:4), T2 (Cu-contaminated soil is mixed with natural soil by 1:2), T3 (Cu-contaminated soil is mixed with natural soil by 1:1) and T4 (Cu-contaminated soil is mixed with natural soil by 1:0).

**Figure 2 plants-09-00404-f002:**
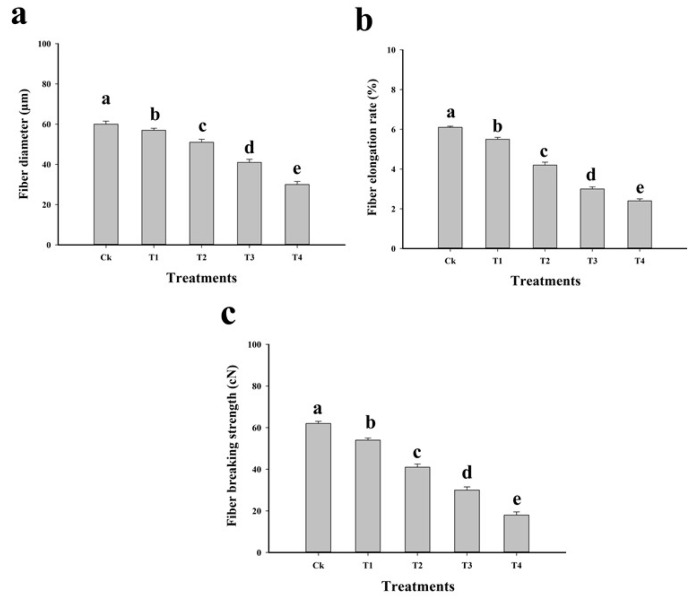
Effect of different levels of Cu-contaminated soil mixed with natural soil on fibre diameter (**a**), fibre elongation rate (**b**) and fibre breaking strength (**c**) of *C*. *capsularis*. Values in the figures are just one harvest. Bars sharing different lowercase letters denote significant statistical differences among the treatments at *p* < 0.05 according to least significant difference (LSD) test. Data shown are the average of three replications (*n* = 3). Ck (soil without addition of Cu-contaminated soil), T1 (Cu-contaminated soil is mixed with natural soil by 1:4), T2 (Cu-contaminated soil is mixed with natural soil by 1:2), T3 (Cu-contaminated soil is mixed with natural soil by 1:1) and T4 (Cu-contaminated soil is mixed with natural soil by 1:0).

**Figure 3 plants-09-00404-f003:**
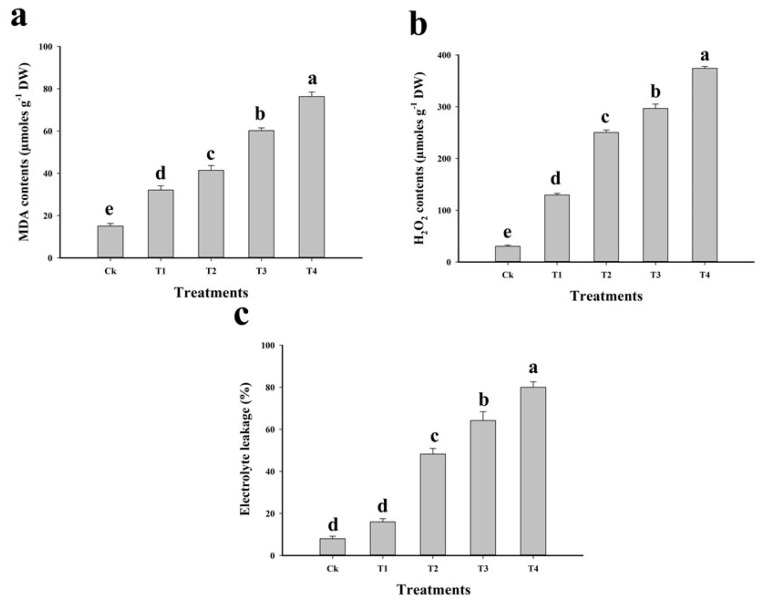
Effect of different levels of Cu-contaminated soil mixed with natural soil on malondialdehyde (MDA) contents (**a**), H_2_O_2_ contents (**b**) and electrolyte leakage (**c**) in the leaves of *C*. *capsularis*. Values in the figures are just one harvest. Bars sharing different lowercase letters denote significant statistical differences among the treatments at *p* < 0.05 according to least significant difference (LSD) test. Data shown are the average of three replications (*n* = 3). Ck (soil without addition of Cu-contaminated soil), T1 (Cu-contaminated soil is mixed with natural soil by 1:4), T2 (Cu-contaminated soil is mixed with natural soil by 1:2), T3 (Cu-contaminated soil is mixed with natural soil by 1:1) and T4 (Cu-contaminated soil is mixed with natural soil by 1:0).

**Figure 4 plants-09-00404-f004:**
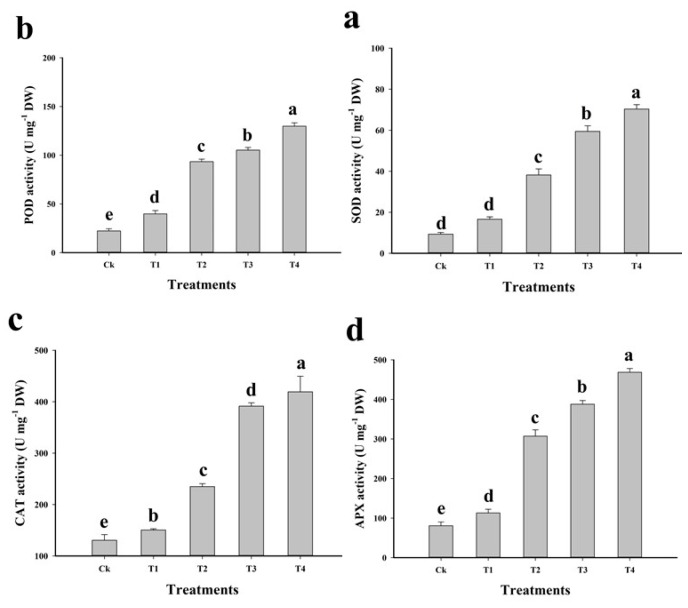
Effect of different levels of Cu-contaminated soil mixed with natural soil on superoxidase dismutase (SOD) (**a**), peroxidase (POD) (**b**), catalase (CAT) (**c**) and ascorbate peroxidase (APX) (**d**) in the leaves of *C. capsularis*. Values in the figures are just one harvest. Bars sharing different lowercase letters denote significant statistical differences among the treatments at *p* < 0.05 according to least significant difference (LSD) test. Ck (soil without addition of Cu-contaminated soil), T1 (Cu-contaminated soil is mixed with natural soil by 1:4), T2 (Cu-contaminated soil is mixed with natural soil by 1:2), T3 (Cu-contaminated soil is mixed with natural soil by 1:1) and T4 (Cu-contaminated soil is mixed with natural soil by 1:0). 2.5. Cu Uptake and Accumulation by Roots and Shoots.

**Figure 5 plants-09-00404-f005:**
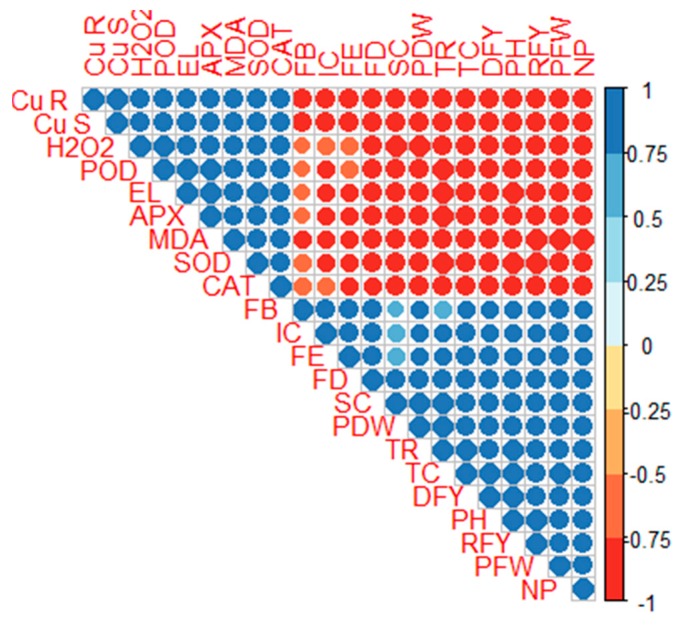
Correlation between Cu uptake and growth attributes of *C. capsularis*. Cu R (Cu concentration in the roots), Cu S (Cu concentration in the shoots), H_2_O_2_ (hydrogen peroxide contents), POD (peroxidase activity), EL (electrolyte leakage), APX (ascorbate peroxidase activity), MDA (malondialdehyde contents), SOD (superoxidase activity), CAT (catalase activity), FB (fibre breaking strength), IC (intercellular CO_2_), FE (fibre elongation rate), FD (fibre diameter), SC (stomatal conductance), PDW (plant dry weight), TR (transpiration rate), TC (total chlorophyll), DFY (degummed fibre yield), PH (plant height), RFY (raw fibre yield), PFW (plant fresh weight) and NP (net photosynthesis).

**Figure 6 plants-09-00404-f006:**
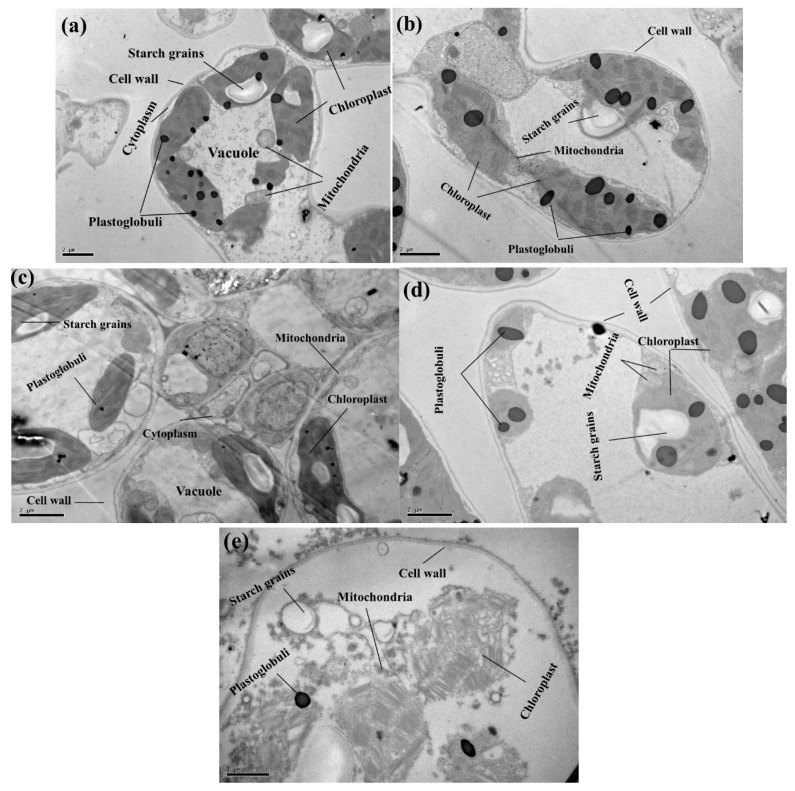
TEM photos of *C. capsularis* leaf cells (**a**) Ck (soil without addition of Cu-contaminated soil) (10,000), (**b**) T1 (Cu-contaminated soil is mixed with natural soil by 1:4) (10,000) (**c**) T2 (Cu-contaminated soil is mixed with natural soil by 1:2) (10,000), (**d**) T3 (Cu-contaminated soil is mixed with natural soil by 1:1) (10,000) and (**e**) T4 (Cu-contaminated soil is mixed with natural soil by 1:0) (5000).

**Table 1 plants-09-00404-t001:** Effect of mixing of two different soils on plant height, plant fresh weight, plant dry weight, total chlorophyll contents, raw fibre yield, and degummed fibre yield on *C. capsularis*.

Treatments	Plant Height (cm)	Plant Fresh Weight (g)	Plant Dry Weight (g)	Total Chlorophyll (mg g^−1^ FW)	Raw Fibre Yield (g)	Degummed Fiber Yield (g)
Ck	241.3 ± 5 a	158.6 ± 4 a	57.6 ± 2 a	2.9 ± 0.01 a	97.0 ± 2 a	25.0 ± 1 a
T1	234.6 ± 4 a	148.0 ± 2 b	51.6 ± 2 b	2.8 ± 0.06 a	92.0 ± 1 a	24.0 ± 1 a
T2	218.3 ± 4 b	138.3 ± 3 c	45.0 ± 1 c	2.4 ± 0.07 b	83.0 ± 2 b	19.0 ± 1 b
T3	198.6 ± 3 c	124.3 ± 3 d	41.0 ± 1 d	2 ± 0.06 c	72.0 ± 3 c	13.6 ± 2 c
T4	184.0 ± 6 d	105.0 ± 3 e	33.6 ± 3 e	1.6 ± 0.08 d	63.3 ± 2 d	9.0 ± 1 d

Values in the table are just one harvest. Different lowercase letters within a column indicate a significant difference between the treatments (*p* < 0.05). Data shown are the average of five replications (*n* = 5). Ck (soil without addition of Cu-contaminated soil), T1 (Cu-contaminated soil is mixed with natural soil by 1:4), T2 (Cu-contaminated soil is mixed with natural soil by 1:2), T3 (Cu-contaminated soil is mixed with natural soil by 1:1) and T4 (Cu-contaminated soil is mixed with natural soil by 1:0).

**Table 2 plants-09-00404-t002:** Cu concentration in roots and shoots of *C. capsularis* grown under different concentration of Cu in the soil.

Treatments	Cu in Roots (mg kg^−1^ DW)	Cu in Shoots (mg kg^−1^ DW)	Shoots/Roots Cu
Ck	14.1 ± 2 e	27.5 ± 3 e	1.9 ± 0.3 e
T1	24.1 ± 2 d	81.6 ± 4 d	3.4 ± 0.3 a
T2	37.5 ± 3 c	115.8 ± 4 c	3.1 ± 0.2 c
T3	55.8 ± 4 b	174.1 ± 4 b	3.2 ± 0.1 b
T4	78.3 ± 4 a	214.1 ± 5 a	2.8 ± 0.1 d

Values in the figures are just one harvest. Bars sharing different lowercase letters denote significant statistical differences among the treatments at *p* < 0.05 according to least significant difference (LSD) test. Ck (soil without addition of Cu-contaminated soil), T1 (Cu-contaminated soil is mixed with natural soil by 1:4), T2 (Cu-contaminated soil is mixed with natural soil by 1:2), T3 (Cu-contaminated soil is mixed with natural soil by 1:1) and T4 (Cu-contaminated soil is mixed with natural soil by 1:0).

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
