# Peer review of "Copper Uptake and Accumulation, Ultra-Structural Alteration, and Bast Fibre Yield and Quality of Fibrous Jute (Corchorus capsularis L.) Plants Grown under Two Different Soils of China"

_plants, 2020, doi:10.3390/plants9030404_

Round 1

Reviewer 1 Report

This is very interesting and promising research.
The conclusions could be more specific.

----------------

Comments for the Review

Title: Copper microlocalisation, ultra-structural alteration, and bast fibre yield and quality of fibrous crop plants grown in copper-contaminated soil in China

Authors: Muhammad Hamzah Saleem , Shafaqat Ali * , Sana Irshad , Muhammad Hussaan , Muhammad Rizwan , Muhammad Shoaib Rana , Abeer Hashem , Elsayed Fathi Abd_Allah , Parvaiz Ahmad *

The research carried out by the Authors on the use of Corchorus capsularis L. for the purification of copper contaminated soils is very interesting and promising. The research carried out by the authors on the use of Corchorus capsularis L. for the purification of copper contaminated soils is very interesting and promising.

The use of natural land purification methods, such as phytoremediation, should be widely used, whenever possible using native native species. The Authors also conducted interesting research on the mechanism of Cu uptake and accumulation in plant tissues and the impact of contamination on its physiological processes.

Research on the use of plants for phytoremediation of pollutants in degraded areas and on "defense" mechanisms of plants is carried out in many countries, including Poland, especially in areas degraded by industry, including the mining industry. The studies described by the authors constitute a significant contribution in this field of knowledge.

The article is written correctly and understandably (it contained minor language shortcomings, but I don't feel qualified to judge about the English language and style)

The authors undertook to extend the conclusions, which in my opinion were too brief in original version (Authors reply to Review Report (round1)

Author Response

Reviewer # 1

Comment # 1: This is very interesting and promising research. The conclusions could be more specific.

Response: Thank you for your suggestions. I have re-written this part of manuscript “conclusion” and used Light Green colour for your response in the manuscript

Reviewer 2 Report

The authors have reported growth, development and yield of fibrous white jute under Cu stress. The manuscript has some points which I am mentioning below

  1. The authors should also measure and mention concentration of some other heavy metal ions in the different soil samples. And it will be better if they could provide more physiochemical properties like carbon, phosphorous, nitrogen content of the soils used, which is the main focus of the study.
  2. The English language needs to be addressed, few of the points mentioned below
  • Rephrase sentence from line number 62-65
  • Rephrase sentence from line number 76-77
  • Rephrase sentence from line number 99-100
  • “bounded” should be bound in line number 118 and 160
  • Change the word “decreased” in line number 126
  • “term” should be terms in line number 138
  • Rephrase sentence from line number 143-146
  • Rephrase sentence from line number 168-169
  • Rephrase sentence from line number 170-171
  • Rephrase sentence from line number 192-194
  • Rephrase sentence from line number 281-282
  • Rephrase sentence from line number 291-292
  • Rephrase sentence from line number 297-298
  • “water” should be watered in line number 300
  •  
  • “weighting” should be weighing in line number 310
  • Rephrase Conclusion section. Avoid use of however in most of the sentences.
  1. In the title it says Cu microlocalization…what does microlocalization means. Does the author mean differential root and shoot localization? Please change accordingly
  2. Give more descriptive heading in each section, such as “growth and Biomass is affected by Cu stress”
  3. Explain Ck, T1, T2 and so on at the start of the results section, so that it is not difficult to understand the experiment
  4. Explain Pn, gs, Ts, and Ci in line number 120.
  5. The figures are blurred and the text in the figures are not readable. Increase the font size and resolution of the figures
  6. Explain the abbreviations, the first time they are used like EL in line number 163, CRD in 285
  7. Why in each of the figures it is written that ‘Values in the “table” are just one harvest’. Please change accordingly.

Author Response

Reviewer # 2

Comment # 1: The authors have reported growth, development and yield of fibrous white jute under Cu stress. The manuscript has some points which I am mentioning below:  The authors should also measure and mention concentration of some other heavy metal ions in the different soil samples. And it will be better if they could provide more physiochemical properties like carbon, phosphorous, nitrogen content of the soils used, which is the main focus of the study.

Response: Respected reviewer, I have mentioned the following information as you mentioned here. Although this experiment is the mixing of two different soils so the properties are very important. Now separate and mixed soil properties are already mentioned in the manuscript.

Comment # 2: The English language needs to be addressed, few of the points mentioned below: Rephrase sentence from line number 62-65

Response: Respected reviewer, I have re-written this sentence.

Comment # 3: Rephrase sentence from line number 76-77.

Response: Respected reviewer, I have re-written this sentence.

Comment # 4: Rephrase sentence from line number 99-100

Response: Respected reviewer, I have re-written this sentence.

Comment # 5: “bounded” should be bound in line number 118 and 160

Response: Respected reviewer, I have improved these mistakes.

Comment # 6: Change the word “decreased” in line number 126

Response: Respected reviewer, I have changed it with “affect”.

Comment # 7: “term” should be terms in line number 138

Response: Respected reviewer, I have improved these mistakes.

Comment # 7: Rephrase sentence from line number 143-146

Response: Respected reviewer, I have re-written this sentence.

Comment # 8: Rephrase sentence from line number 168-169

Response: Respected reviewer, I have re-written this sentence.

Comment # 9: Rephrase sentence from line number 170-171

Response: Respected reviewer, I have re-written this sentence.

Comment # 10: Rephrase sentence from line number 192-194

Response: Respected reviewer, I have re-written this sentence.

Comment # 11: Rephrase sentence from line number 281-282

Response: Respected reviewer, I have re-written this sentence.

Comment # 12: Rephrase sentence from line number 291-292

Response: Respected reviewer, I have re-written this sentence.

Comment # 13: Rephrase sentence from line number 297-298

Response: Respected reviewer, I have re-written this sentence.

Comment # 14: “water” should be watered in line number 300

Response: Respected reviewer, I have improved these mistakes.

 Comment # 15: “weighting” should be weighing in line number 310

Response: Respected reviewer, I have improved these mistakes.

Comment # 16: Rephrase Conclusion section. Avoid use of however in most of the sentences.

Response: Respected reviewer, I have improved these mistakes.

Comment # 17: In the title it says Cu microlocalization…what does microlocalization means. Does the author mean differential root and shoot localization? Please change accordingly

Response: Respected reviewer, as the meaning of microlocalization was not cleared so I replaced it with “uptake and accumulation” and changed whole title which is more relevant to the present manuscript.

Comment # 18: Give more descriptive heading in each section, such as “growth and Biomass is affected by Cu stress”

Response: Respected reviewer, I have changed this heading.

Comment # 19: Explain Ck, T1, T2 and so on at the start of the results section, so that it is not difficult to understand the experiment

Response: Respected reviewer, thanks for your suggestions. I have written this detail in the start of this section.

Comment # 20: Explain Pn, gs, Ts, and Ci in line number 120.

Response: Respected reviewer, I write these again in the manuscript as you mentioned.

Comment # 21: The figures are blurred and the text in the figures are not readable. Increase the font size and resolution of the figures

Response: Respected reviewer, I have made new graphs in Sigmaplot with 600 mpi.

Comment # 22: Explain the abbreviations, the first time they are used like EL in line number 163, CRD in 285

Response: Respected reviewer, I have improved these mistakes.

Comment # 23: Why in each of the figures it is written that ‘Values in the “table” are just one harvest’. Please change accordingly.

Response: Respected reviewer, I have changed accordingly and I also attached English Editing Certificate with revised manuscript. Thank you for your suggestions and time and I used Red Color in the manuscript to response your comments.

Reviewer 3 Report

The MS shows interesting data, but there are some unclear parts, some errors and inaccuracies; therefore, a revision is necessary.

In fact:

  • An accurate English check, is necessary (e.g. see lines 39-40: “Furthermore, Cu was highly transported to the harvestable parts than in belowground parts while it was also noticed that increasing Cu contents in the soil, increased Cu uptake by plant tissues”);
  • Table 1: the units must be indicate at the top of the column for each value, not in the Legend, e.g. Plant Height (cm);
  • Tables and Figures: the phrase “Values in the table (figure) is just one harvest” must be corrected or removed;
  • Table 1: the plant fresh weight values are practically is identical to the raw fibre yield, please check because fibre is not water neither leaves or roots;
  • Figure 1-4: part (b) should be positioned on the right of (a), then (c) below (a), and (d) on the right of (c); the correct order was used for Fig. 6;
  • Sample names: instead of T1, T2, T3 and T4 that correspond to decreasing levels of Cu in the soil, Authors should use the Cu levels starting from the lower one: 287, 452, 967 and 1719 (mg kg-1 Cu); in tis way the x axis of Fig. 1-4 will indicate increasing values of Cu as logic; furthermore, it is appropriate to call the different samples with the concentration of copper in the soil to facilitate reading;
  • Also, the fonts included in the figures are too small, please increase their size;
  • Figures 3 and 4: values should be expressed for dry weight;
  • Table 2: it is not indicated if the values are for DW or FW (DW must be preferred); furthermore Authors must clearly define “shoot”: stems + leaves or only stem?
  • Table 2: Cu concentration in Ck shoot is 28 mg kg−1; Authors should comment on the fact that this value is very close to the Cu content of the soil control (34 mg kg−1);
  • Figure 6: please use an identical magnification and use bigger character on the size bars;
  • In M&M: it is necessary to indicate the common name of the plant together with the Latin name, white jute / Corchorus capsularis;
  • Line 264: please explain what are the “controlled glass conditions”;
  • Line 272: the statement “Ck (soil without Cu concentration)” is not true because at line 279 the Cu concentration in Ck is 34 mg kg−1.

Author Response

Reviewer # 3

Comment # 1: The MS shows interesting data, but there are some unclear parts, some errors and inaccuracies; therefore, a revision is necessary. In fact: An accurate English check, is necessary (e.g. see lines 39-40: “Furthermore, Cu was highly transported to the harvestable parts than in belowground parts while it was also noticed that increasing Cu contents in the soil, increased Cu uptake by plant tissues”);

Response: Respected reviewer, I have re-written this sentence.

Comment # 2: Table 1: the units must be indicate at the top of the column for each value, not in the Legend, e.g. Plant Height (cm);

Response: Respected reviewer, I have changed according to your suggestions.

Comment # 3: Tables and Figures: the phrase “Values in the table (figure) is just one harvest” must be corrected or removed;

Response: Respected reviewer, also suggested by reviewer#2 and I have corrected this sentence.

Comment # 4: Table 1: the plant fresh weight values are practically is identical to the raw fibre yield, please check because fibre is not water neither leaves or roots;

Response: Respected reviewer, thank you for your suggestions. I have checked again all my raw data and verified my results and write accordingly.

Comment # 5: Figure 1-4: part (b) should be positioned on the right of (a), then (c) below (a), and (d) on the right of (c); the correct order was used for Fig. 6;

Response: Respected reviewer, I have made new graphs by using SigmaPlot Software.

Comment # 6: Sample names: instead of T1, T2, T3 and T4 that correspond to decreasing levels of Cu in the soil, Authors should use the Cu levels starting from the lower one: 287, 452, 967 and 1719 (mg kg-1 Cu); in tis way the x axis of Fig. 1-4 will indicate increasing values of Cu as logic; furthermore, it is appropriate to call the different samples with the concentration of copper in the soil to facilitate reading;

Response: Respected reviewer, to avoid any confusion I have written the detailed of my treatments in the start of result section and at the end of all presented data such as tables and figures etc. My main focus to use such same names of the treatments as this experiment is the part of my thesis and I already used these same treatments in a flax experiment which I mentioned in the Materials And Methods. https://doi.org/10.1007/s11356-019-07264-7.

Comment # 7: Also, the fonts included in the figures are too small, please increase their size;

Response: Respected reviewer, I have made new graphs according to your suggestions.

Comment # 8: Figures 3 and 4: values should be expressed for dry weight;

Response: Respected reviewer, as this is enzymatic study and we used fresh weight data to calculate these results so that’s why we used “FW” in these figures.

Comment # 9: Table 2: it is not indicated if the values are for DW or FW (DW must be preferred); furthermore, Authors must clearly define “shoot”: stems + leaves or only stem?

Response: Respected reviewer, I have pointed out these mistakes in the text and shoot means leaves, stems and fibres.

Comment # 10: Table 2: Cu concentration in Ck shoot is 28 mg kg−1; Authors should comment on the fact that this value is very close to the Cu content of the soil control (34 mg kg−1);

Response: Respected reviewer, Cu is an micronutrient and little Cu was also accumulated by jute from the soil so that’s why Cu in control is very close of Cu in the soil. And Also its 120 days experiment so that is also a reason which we have proved in detail in our previous literature. https://doi.org/10.1016/j.ecoenv.2019.109915.

Comment # 11: Figure 6: please use an identical magnification and use bigger character on the size bars;

Response: Respected reviewer, I have re-write these bars again with bigger size and magnification is clearer now.

Comment # 12: In M&M: it is necessary to indicate the common name of the plant together with the Latin name, white jute / Corchorus capsularis;

Response: Respected reviewer, I have written all available information regarding jute in M&M.

Comment # 13: Line 264: please explain what are the “controlled glass conditions”;

Response: Respected reviewer, I have replaced it with “glass house environment under controlled conditions”.

Comment # 14: Line 272: the statement “Ck (soil without Cu concentration)” is not true because at line 279 the Cu concentration in Ck is 34 mg kg−1.

Response: Respected reviewer, thanks for your time and suggestions. Although terminology is not true to write in this way but where our means is soil without addition of Cu Contaminated soil and we used this throughout the text. But now I changed it and for the suggestions of your comments I used Blue color in the MS.

Round 2

Reviewer 2 Report

The authors have improved the manuscript considerably. Though the authors have provided English editing certificate, I would like to mention some minor additional things

  1. Please change the title to “Copper Uptake and Accumulation, Ultra-Structural Alteration, Bast Fibre Yield and Quality of Fibrous Jute (Corchorus capsularis L.) Plants Grown Under Two Different Soils of China”
  2. Please change the titles in all the sections, self-descriptive titles in each section, not only in one section which I mentioned earlier
  3. In line number 43 there should not be the word “of”. In the same sentence was is in italics
  4. Line number 84, there should be “discussed in detail”
  5. Line 424, change “grown” to “can grow”
  6. And I still think resolution of the graphs can be increased

Author Response

Comment # 1: The authors have improved the manuscript considerably. Though the authors have provided English editing certificate, I would like to mention some minor additional things: Please change the title to “Copper Uptake and Accumulation, Ultra-Structural Alteration, Bast Fibre Yield and Quality of Fibrous Jute (Corchorus capsularis L.) Plants Grown Under Two Different Soils of China”

Response: Respected reviewer, I have made these followings changes in the title.

Comment # 2: Please change the titles in all the sections, self-descriptive titles in each section, not only in one section which I mentioned earlier

Response: Respected reviewer, I have made these changes accordingly. Thanks for your suggestions,

Comment # 3: In line number 43 there should not be the word “of”. In the same sentence was is in italics

Response: Respected reviewer, I have made this change in MS.

Comment # 4: Line number 84, there should be “discussed in detail”

Response: Respected reviewer, I have made this change in MS.

Comment # 5: Line 424, change “grown” to “can grow”

Response: Respected reviewer, I have made this change in MS.

Comment # 6: And I still think resolution of the graphs can be increased

Response: Respected reviewer, I have made again new graphs with SigmaPlot and for the response of your comments I used Blue color.

Reviewer 3 Report

The MS was improved but still some errors and inaccuracies are present whereas the Authors' response is unsatisfactory; therefore, a major revision is necessary.

In fact:

Comment # 2: Table 1: the units must be indicate at the top of the column for each value, not in the Legend, e.g. Plant Height (cm);

Response: Respected reviewer, I have changed according to your suggestions.

In table 1 the units for “Raw FibreYield (g)” and “Degummed Fiber Yield (g)” are not clear: is that for plant or for pot? Expressed as DW or FW? Moreover, the values of “Raw FibreYield (g)” were changed from the previous version without a comment. Finally, I believe that all values must be expressed as DW which represent really the biomass produced.

Comment # 4: Table 1: the plant fresh weight values are practically is identical to the raw fibre yield, please check because fibre is not water neither leaves or roots;

Response: Respected reviewer, thank you for your suggestions. I have checked again all my raw data and verified my results and write accordingly.

See above and please clarify what is “raw fibre” in M&M. Also, if the fibres are expressed as DW the values are not correct, if the fibres are expressed as FW the values are unlikely because the plant also has leaves and roots…..; finally Authors cited their previous paper of Saleem et al. [49] Ecotoxicology and Environmental Safety Volume 189, February 2020 (not 2019!!!), 109915 without considering that in such paper the Cu content in roots and shoots is extremely high (see Fig. 3 of Saleem et al.) and total chlorophyll content is low (see Table 1) as two treatments are close to T3 and T4.

Comment # 6: Sample names: instead of T1, T2, T3 and T4 that correspond to decreasing levels of Cu in the soil, Authors should use the Cu levels starting from the lower one: 287, 452, 967 and 1719 (mg kg-1 Cu); in tis way the x axis of Fig. 1-4 will indicate increasing values of Cu as logic; furthermore, it is appropriate to call the different samples with the concentration of copper in the soil to facilitate reading;

Response: Respected reviewer, to avoid any confusion I have written the detailed of my treatments in the start of result section and at the end of all presented data such as tables and figures etc. My main focus to use such same names of the treatments as this experiment is the part of my thesis and I already used these same treatments in a flax experiment which I mentioned in the Materials And Methods. https://doi.org/10.1007/s11356-019-07264-7.

Again, because normally results are showed starting from a lower treatment value, Authors must at least indicate T1 the Cu contaminated soil mixed with natural soil by 1:4 ), T2 the Cu contaminated soil mixed with natural soil by 1:2, T3 the mix 1:1, and T4 the mix 0:1.

Comment # 8: Figures 3 and 4: values should be expressed for dry weight;

Response: Respected reviewer, as this is enzymatic study and we used fresh weight data to calculate these results so that’s why we used “FW” in these figures.

Sorry, but for Figure 3 the values are chemical compounds, whereas for Figure 4 the correct expression is Enzymatic Units / mg protein. So, please use the above mentioned units, and use DW throughout the MS.

Comment # 9: Table 2: it is not indicated if the values are for DW or FW (DW must be preferred); furthermore, Authors must clearly define “shoot”: stems + leaves or only stem?

Response: Respected reviewer, I have pointed out these mistakes in the text and shoot means leaves, stems and fibres.

Ok, but now a Cu ratio shoots / roots is necessary because the higher Cu accumulation is for the treatment Cu contaminated soil plus natural soil by 1:4.

Comment # 10: Table 2: Cu concentration in Ck shoot is 28 mg kg−1; Authors should comment on the fact that this value is very close to the Cu content of the soil control (34 mg kg−1);

Response: Respected reviewer, Cu is an micronutrient and little Cu was also accumulated by jute from the soil so that’s why Cu in control is very close of Cu in the soil. And Also its 120 days experiment so that is also a reason which we have proved in detail in our previous literature. https://doi.org/10.1016/j.ecoenv.2019.109915.

Authors are invited to discuss their actual results with the previous cited article of Saleem et al. 2020. In fact, it seems to me very strange that the Authors themselves, growing C. capsularis, use for a work precise concentrations of Cu, (0, 100, 200, 300, and 400 mg kg−1), for the following mixes of Cu polluted and unpolluted soils ( from 1:0 to 1:4).

Finally, in References, all species’ names must bi in Italics

Author Response

Comment # 1: The MS was improved but still some errors and inaccuracies are present whereas the Authors' response is unsatisfactory; therefore, a major revision is necessary. In fact: Comment # 2: Table 1: the units must be indicating at the top of the column for each value, not in the Legend, e.g. Plant Height (cm); Response: Respected reviewer, I have changed according to your suggestions.

In table 1 the units for “Raw Fibre Yield (g)” and “Degummed Fiber Yield (g)” are not clear: is that for plant or for pot? Expressed as DW or FW? Moreover, the values of “Raw Fibre Yield (g)” were changed from the previous version without a comment. Finally, I believe that all values must be expressed as DW which represent really the biomass produced.

Response: Respected reviewer, I have checked my Raw data and there was a mistake in my means value and I changed according to my raw data. Although, your suggestion is quite right about the values of FW and Raw Fibre Yield. The values mention in Raw Fibre Yield and Degummed Fiber Yield are the sum or mean of three values and we expressed in the grams which also previously reported by Sana et al. DOI:10.1038/s41598-017-09584-5 and doi:10.3390/su8090887.

 As also shown in the figure. The units are for the plants and the values for Raw Fibre Yield are belongs to FW while for Degummed Fibre yield are for DW. to avoid any confusion I also mention in the M&M. Although your suggestions are quite right all the values (except fresh weight of the plant) are in DW.

Comment # 2: Comment # 4: Table 1: the plant fresh weight values are practically is identical to the raw fibre yield, please check because fibre is not water neither leaves or roots; Response: Respected reviewer, thank you for your suggestions. I have checked again all my raw data and verified my results and write accordingly. See above and please clarify what is “raw fibre” in M&M. Also, if the fibres are expressed as DW the values are not correct, if the fibres are expressed as FW the values are unlikely because the plant also has leaves and roots…..; finally Authors cited their previous paper of Saleem et al. [49] Ecotoxicology and Environmental Safety Volume 189, February 2020 (not 2019!!!), 109915 without considering that in such paper the Cu content in roots and shoots is extremely high (see Fig. 3 of Saleem et al.) and total chlorophyll content is low (see Table 1) as two treatments are close to T3 and T4.

Response: Respected reviewer, I have highlighted Raw Fibre Yield in Tables and also mentioned in the M&M “The fibre layer of each stem was decorticated (peeled from the pith), the epidermis was removed, and raw fibres were weighed for calculation of fibre yield”. The values for Raw Fibre Yield are belongs to FW while for Degummed Fibre yield are for DW. Suppose for T1 Total fresh weight was 105g and Raw Fibre yield was 64g. This is because that jute plants in its maturity stage lost most of the leaves and we can say just stem was remained.

These are just rough photos and can see the difference between first photo and the last one. However, in our previous paper we did not took fibrous parameters such as Raw Fibre yield and Degummed Fibre yield. And you are asking about Cu concentration in the roots and shoots and Total chlorophyll contents. This is because the soil used in the present study was collected from a Cu mining area of Baisha Village, DaYe County, Hubei, China (115.20′E, 29.85′N) at a depth of 0–25 cm. As mentioned in the manuscript. And In our previous paper we used normal soil with artificially spiked Cu. This is the main reason the Cu contents was higher in previous study and also soil properties are also take part in it. Acidic soil helps is accumulation of metal However in the present study pH of T1 was 6.9 and in the previous study pH was 5.8. However, chlorophyll contents and other morphological traits and in previous study was reduced under Cu toxicity at 400 mg/kg Cu while in the present study Cu toxicity was less because of the soil properties as mentioned earlier.

Comment # 3: Comment # 6: Sample names: instead of T1, T2, T3 and T4 that correspond to decreasing levels of Cu in the soil, Authors should use the Cu levels starting from the lower one: 287, 452, 967 and 1719 (mg kg-1 Cu); in this way the x axis of Fig. 1-4 will indicate increasing values of Cu as logic; furthermore, it is appropriate to call the different samples with the concentration of copper in the soil to facilitate reading; Response: Respected reviewer, to avoid any confusion I have written the detailed of my treatments in the start of result section and at the end of all presented data such as tables and figures etc. My main focus to use such same names of the treatments as this experiment is the part of my thesis and I already used these same treatments in a flax experiment which I mentioned in the Materials And Methods. https://doi.org/10.1007/s11356-019-07264-7.

Again, because normally results are showed starting from a lower treatment value, Authors must at least indicate T1 the Cu contaminated soil mixed with natural soil by 1:4 ), T2 the Cu contaminated soil mixed with natural soil by 1:2, T3 the mix 1:1, and T4 the mix 0:1.

 Response: Respected reviewer, thanks for your suggestion. According to your instructions I have made these changes i.e. make graphs and tables again and check these treatments throughout the manuscript. Now it will be easy for the readers to understand my concept.

Comment # 4: Comment # 8: Figures 3 and 4: values should be expressed for dry weight; Response: Respected reviewer, as this is enzymatic study and we used fresh weight data to calculate these results so that’s why we used “FW” in these figures. Sorry, but for Figure 3 the values are chemical compounds, whereas for Figure 4 the correct expression is Enzymatic Units / mg protein. So, please use the above mentioned units, and use DW throughout the MS.

Response: Respected reviewer, thanks for your suggestion I have made these changes in the graphs and MS also.

Comment # 5: Comment # 9: Table 2: it is not indicated if the values are for DW or FW (DW must be preferred); furthermore, Authors must clearly define “shoot”: stems + leaves or only stem?

Response: Respected reviewer, I have pointed out these mistakes in the text and shoot means leaves, stems and fibres.

Ok, but now a Cu ratio shoots / roots is necessary because the higher Cu accumulation is for the treatment Cu contaminated soil plus natural soil by 1:4.

 Response: Respected reviewer, I have measured the ratio between Cu contents in the shoots to the roots and mentioned in the Table 2. Thanks for your suggestions.

Comment # 6: Comment # 10: Table 2: Cu concentration in Ck shoot is 28 mg kg−1; Authors should comment on the fact that this value is very close to the Cu content of the soil control (34 mg kg−1);

Response: Respected reviewer, Cu is an micronutrient and little Cu was also accumulated by jute from the soil so that’s why Cu in control is very close of Cu in the soil. And Also its 120 days experiment so that is also a reason which we have proved in detail in our previous literature. https://doi.org/10.1016/j.ecoenv.2019.109915. Authors are invited to discuss their actual results with the previous cited article of Saleem et al. 2020. In fact, it seems to me very strange that the Authors themselves, growing C. capsularis, use for a work precise concentrations of Cu, (0, 100, 200, 300, and 400 mg kg−1), for the following mixes of Cu polluted and unpolluted soils ( from 1:0 to 1:4).

Response: Respected reviewer, As these studies are the part of my thesis and even I already mentioned in M&M that these treatments are already used in my previous study of the flax. And I already mentioned that the difference between the results of two same study is because of the soil properties. As this soil was wild soil or mining area’s soil so there is a give difference in the properties of both soil and I already mentioned that morphology and Cu accumulation is greatly depend upon the soil properties so that’s why there is a big difference in the results of these two studies.

Comment # 7: Finally, in References, all species’ names must bi in Italics

Response: Respected reviewer, references I set with the endnote software and the response for your comments I used red color.

Round 3

Reviewer 3 Report

Finally the MS has been really improved and only minor changes remain to be made:
. make English fluent;
. add the missing decimals in Tables 1 and 2, so to have the same number of representative digits (example, Tab. 1, add a zero after 2 to have 2.0±0.06c);
. add a comment in the text about the trend of the shoot/root ratio for copper content (Table 2).

Author Response

Comment # 1: Finally the MS has been really improved and only minor changes remain to be made: make English fluent; add the missing decimals in Tables 1 and 2, so to have the same number of representative digits (example, Tab. 1, add a zero after 2 to have 2.0±0.06c);

Response: Respected reviewer, I have made these followings changes in all the tables and also pointed out my mistakes in the MS.

Comment # 2: add a comment in the text about the trend of the shoot/root ratio for copper content (Table 2).

Response: Respected reviewer, I have made these followings changes in the MS and marked as Red.